# Spatial-Temporal Change for Ecological Intactness of Giant Panda National Park and Its Adjacent Areas in Sichuan Province, China

**Chuan Luo** [1,2]**, Hao Yang** [1,]*****, Peng Luo** [1,]*****, Shiliang Liu** [3,]*****, Jun Wang** [4]**, Xu Wang** [5]**, Honglin Li** [1,2]**,
Chengxiang Mou** [6]**, Li Mo** [7]**, Honghong Jia** [1,2]**, Sujuan Wu** [1,2]**, Yue Cheng** [1,2]**, Yu Huang** [1,2]** and Wenwen Xie** [1,2]

1 CAS Key Laboratory of Mountain Ecological Restoration and Bioresource Utilization & Ecological Restoration and Biodiversity Conservation Key Laboratory of Sichuan Province, Chengdu Institute of Biology, Chinese Academy of Sciences, Chengdu 610041, China; luochuan@cib.ac.cn (C.L.); lihl@cib.ac.cn (H.L.); jiahh@cib.ac.cn (H.J.); wusujuan18@mails.ucas.ac.cn (S.W.); chengyue20@mails.ucas.ac.cn (Y.C.); huangyu21@mails.ucas.ac.cn (Y.H.); xiewenwen21@mails.ucas.ac.cn (W.X.)
2 University of Chinese Academy of Sciences, Beijing 100049, China
3 State Key Laboratory of Water Environment Simulation, School of Environment, Beijing Normal University, Beijing 100875, China
4 College of Environmental Science and Engineering, China West Normal University, Nanchong 637000, China; drjunw@163.com
5 Ya'an Branch of Giant Panda National Park Administration, Ya'an 625000, China; wangxuexu888.student@sina.com
6 College of Chemistry and Life Sciences, Sichuan Provincial Key Laboratory for Development and Utilization of Characteristic Horticultural Biological Resources, Chengdu Normal University, Chengdu 611130, China; mouchengxiang@126.com
7 Sichuan Key Laboratory of Conservation Biology for Endangered Wildlife, Chengdu Research Base of Giant Panda Breeding, Chengdu 610081, China; jasmineae@126.com
* Correspondence: yanghao@cib.ac.cn (H.Y.); luopeng@cib.ac.cn (P.L.); shiliangliu@bnu.edu.cn (S.L.)

**Abstract:** Human activities change the natural ecosystem and cause the decline of the intact ecosystem. Establishing an applicable and efficient human activity monitoring indicator system benefits China's ambitious national park system construction. In this study, we established a refined technique for ecological intactness scores (EIS) and applied it in the area of Giant Panda National Park (GPNP) from 1980 to 2020 by quantifying four types of human interferences including land use and cover change (LUCC), road construction, water reservoir and hydropower construction, and mining. The results show the following: (1) Under the ecological intactness score range of 0–10, the GPNP with about 92.6% area of the EIS was above 6.0, and the mean baseline level of intactness was 7.1 when it was established in 2018. (2) The EIS in the east of Qionglaishan and south of Minshan were relatively lower than the rest of the study area. (3) During the past 40 years, 80% of the GPNP's ecological intactness has remained stable. (4) In total, 14% of the GPNP was degraded mainly in the areas below 1200 m with severe human activities. (5) LUCC and road construction were the main driving factors for the decrease of ecological intactness in the GPNP. (6) The habitat of the giant panda is mainly distributed in the areas with an EIS above 6.0, and this is a key link between ecological intactness and habitat suitability. Our research proved that the ecological intactness score (EIS) is an effective indicator for monitoring and assessing the impact of human activities on the regional natural ecosystem and could be helpful for ecological restoration and human activities management GPNP in the future.

**Keywords:** national park; ecological intactness; human activity; land use and cover change; habitat suitability

## 1. Introduction

China has launched an ambitious plan to build a national park system to protect its most important natural ecosystems and their associated biodiversity in response to the current climate change risks and biodiversity crises at the regional and global scales. The newly established Giant Panda National Park (GPNP) and its adjacent areas are one of the hotspots for global biodiversity conservation [1] and the priority areas for biodiversity conservation in China [2]. Many studies have revealed that human disturbance is one of the main reasons for the fragmentation and degradation of giant panda habitats [3–5]. Conservation of the giant panda's habitat and maintenance of its intact ecosystem have been set as the core goals for the new GPNP. However, a report about the regional ecosystem status or comprehensive effects of human disturbance on giant panda habitat is rare [5]. Selecting appropriate and efficient evaluation protocols and indicators is the key issue for achieving conservation management goals [6].

With the widespread of nature conservation practices, monitoring and evaluation systems for conservation have been continuously enriched. Representative evaluation tools include the following: intactness evaluation [7,8], naturalness evaluation [9,10], and integrity evaluation [11,12]. In the broad field of conservation biology "intactness"," naturalness", and "integrity" can be treated as synonymous [10,13]. From our point of view, these three indicator systems have their own focus: (1) naturalness, commonly defined as the similarity of a current ecosystem state to its natural state, focuses on "compositional", "structural", and "functional" indicators of the ecosystem [10]; (2) intactness is the state of an ecosystem that develops in response to natural processes without human interference, with an emphasis on measuring the degree of human disturbance to natural ecosystems [7]; (3) ecosystem integrity is a huge notion with meanings ranging from intactness, wholeness, health, and functioning to quality and resilience, which pays attention to the absence of ecosystem elements (wholeness) [6,14,15]. Naturalness, intactness, and ecosystem integrity are continuity concepts for supporting ecosystem and biodiversity conservation aims of sustainable development [10,16]. Based on our knowledge, we chose the term "intactness" to construct an indicator that can measure the effectiveness of national parks in managing human activities at the landscape scale.

In this study, we define the degree to which an ecosystem retains its natural state unmodified by remote sensing identifiable human activity as ecological intactness. Traditionally, China and other countries have adopted forest management practices, such as improvement of forest cover, tree stock volume, and carbon storage as the goals of natural ecosystem preservation [17,18]. These forest management goals may ignore the benefits of ecosystem services and natural values other than wood fiber production [19], such as biodiversity conservation, water regulation, water retention, and climate change mitigation. Many studies have shown that an intact ecosystem supports significant environmental values including biodiversity [20,21], carbon sinks [22], water regulation and provision [23,24], and the maintenance of human health [25]. Hence, ecological intactness could be one valuable indicator for the management of GPNP to reflect human activity and the intact degree of an ecosystem.

The assessment of ecological intactness requires the measurement of human activities and their caused ecosystem changes. Methods to quantify human activity in an ecosystem include human footprint [26,27] and human modification [28,29]. Those methods collect human activity indicators including building, croplands, pasture lands, nighttime lights, railways, etc. to assess human modification or footprint in continental-scale and larger regional areas. Those studies revealed that 60% of all land changes are associated with direct human activities, and the protection of the intact ecosystem is needed [26,28,30–34]. Human activities are not mutually exclusive and usually co-occur in the same location [35]. The regional assessment of anthropogenic impact is generally conducted by remote sensing [36]. The critical step for assessing the ecological intactness is to identify the main disturbance factors in Giant Panda National Park, and the identifiability of the image is an important criterion for screening the types of human activities involved in the evaluation

of ecological intactness. Land Use and Cover Change (LUCC) is an important material for regional ecological intactness assessment, which contains basic anthropogenic activity information such as villages, cultivated land, etc. According to the evaluation of the 3rd and 4th National Panda Surveys [37–39] and the World Natural Heritage Outlook Report of Giant Panda Habitat [40], road interference, mining interference, water reservoirs, and hydropower construction are identified as the major human disturbances affecting the regional ecological environment.

Previous work aimed to protect the quality of the giant panda habitat. The indicators for evaluating and monitoring giant pandas' habitat quality came from the habitat suitability index (HSI), which is based on environmental factors including vegetation types, elevation, slope, and bamboo distribution [37,41,42]. While the new GPNP takes ecological intactness as a conservation goal in management, it is interesting and important to test if the "intactness" can cover the spatial area of "suitable habitat". With these regards, this study focuses on the following aims: (1) studying the current status of the ecological intactness so as to provide a baseline for GPNP, (2) detecting the changes of ecological intactness in GPNP during the past 40 years and identifying specific areas in need of management, and (3) testing the relationship between ecological intactness and the quality of the panda habitat.

## 2. Materials and Methods

### 2.1. Study Area

The Giant Panda National Park lies in the mountainous area of southwest China, spanning the Sichuan, Gansu, and Shanxi Provinces. We considered all counties of Sichuan Province that are partially or fully designated as national parks as our study area (Figure 1). Sichuan has 1387 giant panda individuals accounting for 74.4% of the total population, and the GPNP Sichuan part composed 74.6% of the total GPNP. The study area can be divided into two parts, one is called GPNP, and the other is the study area outside the GPNP called the exterior area (EA). The research area covered 52,473.3 km$^2$ and the GPNP occupied 20,177 km$^2$ (Figure 1). This study area occupies the QionglaiShan, Minshan, Qinling, and Daxiangling mountain regions. The elevation of the study area ranges from 421 m and 6132 m, and the northwest is higher than the southeast.

The Sichuan GPNP involves 119 townships and 20 counties that belong to 7 prefectures and has a population of 89,900 inhabitants. The local economic structure is relatively simple, with mining, hydropower, and other resource development enterprises generating industrial revenue. The main source of income for community residents is agriculture, and some residents are also engaged in mining and processing labor. According to the Fourth National Giant Panda Census, the common disturbances affecting wild giant pandas and their habitat are livestock, roads, and farming (cultivation of row crops), all of which are significantly increasing, while disturbance from logging has been significantly decreasing after the implementation of the Natural Forest Conservation Program (NFCP) and the Grain-to-Green Program (GTGP) [37].

### 2.2. Materials and Method

#### 2.2.1. Data Preprocessing

Taking the regional characteristics of GPNP and data availability into consideration, we finally tailored four categories of human activity (LUCC, roads, mining, and water reservoirs and hydropower construction) and employed the "increasing" fuzzy sum function to map the ecological intactness in a spatially explicit way [30]. The data timing, resolution, and source information of the four categories of human activity used in this study are shown in Table 1. Based on the local studies concerning the effects of roads on the distribution of giant pandas [43,44] and expert judgment, we set the maximum impact range as 3 km and established four different widths of road influence radius, which were 100 m, 500 m, 1000 m, and 3000 m for assigning different human interference scores(Table 2.).

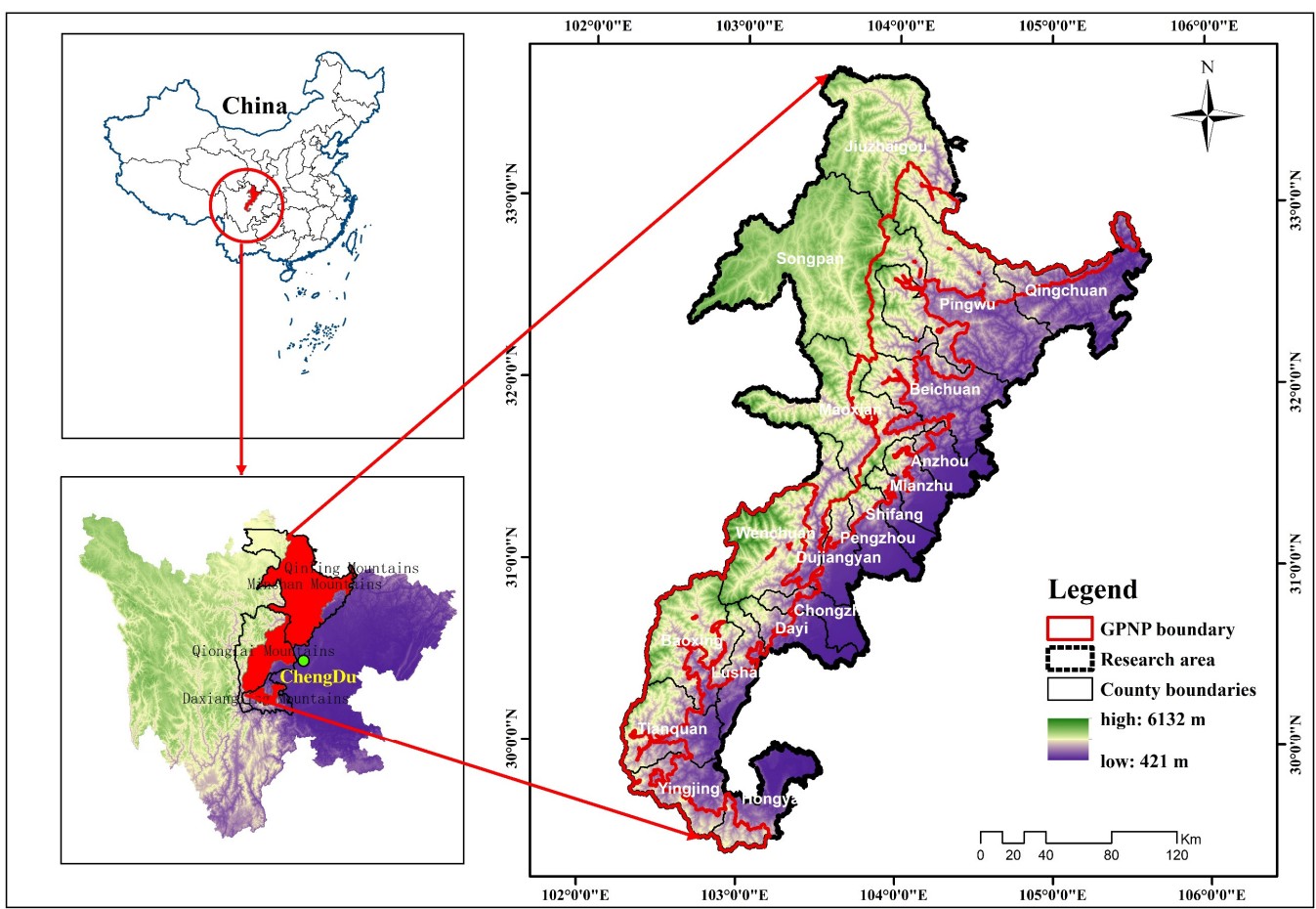

**Figure 1.** Research area and the Giant Panda National Park boundary.

**Table 1.** Overview of human activity, spatial resolution, and the years of data used in the calculation of ecological intactness.

| Human Activity | Year | Resolution | Data Source |
|---|---|---|---|
| Road interference | 1980, 1990, 2000, 2010, 2018, and 2020 | - | The Data Center of Resources and Environmental Sciences, Chinese Academy of Sciences (https://www.resdc.cn/data.aspx?DATAID = 237) (accessed on 6 September 2020); the Traffic Yearbook of Sichuan Province (http://cnki.nbsti.net/CSYDMirror/Trade/yearbook/single/N2021080079?z=Z014) (accessed on 6 September 2020) |
| Mining interference | 1980, 1990, 2000, 2010, 2018, and 2020 | 30 m × 30 m | local authorities; field investigation |
| Water reservoirs and hydropower construction | 1980, 1990, 2000, 2010, 2018, and 2020 | 30 m × 30 m | local authorities; field investigation |
| Land Use and Cover Change | 1980, 1990, 2000, 2010, 2018, and 2020 | 30 m × 30 m | The Data Center of Resources and Environmental Sciences, Chinese Academy of Sciences (https://www.resdc.cn) (accessed on 31 December 2020) |

Mine development data are not available online. We collected the data regarding mine development from local authorities. Those data included the location, area, and times of mining activity. We sorted out six mine development sub-datasets for 1980, 1990, 2000, 2010, 2018, and 2020 according to the date of initial activity. Then, the 2020 sub-dataset was validated with field investigation.

**Table 2.** Road grade * influences degree and scope parameter based on the previous study [30,31] and modified according to related research [43–45] and expert judgment.

| Road Grade | Road Facility (0–100 m) | 100–500 m | 500–1000 m | 1000–3000 m |
|---|---|---|---|---|
| Highway | 10 | 3 | 2 | 1 |
| First Grade | 10 | 3 | 2 | 1 |
| Second grade | 8 | 2 | 1 | 0 |
| Third grade | 6 | 2 | 0 | 0 |
| Fourth grade | 4 | 1 | 0 | 0 |
| Railway | 10 | 3 | 1 | 0 |

* Note: The road grade standards refer to "Road Engineering Technical Standards" (JTGB01-2014).

Human influence scores for mining interference were determined from relevant studies [30,46] and based on onsite investigations. We adjusted the scope of mining interference from 0.5 km to 1 km, and the impact score was adjusted according to our field evaluations on the degree of interference for different mining methods and mining sizes. Mining areas exceeding 1 km$^2$ are considered large, and mining areas less than 1 km$^2$ are considered small. Because mining areas tend to destroy local vegetation and cause slope erosions, mining areas yield the largest impact score of 10, and the impacts gradually fade to zero with increasing distance to 800 m or 1 km. The impact range of mining interference was set at 200 m, 500 m, 800 m, and 1000 m beyond the boundary of the mining area to establish buffers relative to what was being impacted (Table 3).

**Table 3.** Parameter table of influence degree and scope of mining type determined from relevant studies [30,46], and expert judgment based on onsite investigations.

| Mining Type | Mining Area | 0–200 m | 200–500 m | 500–800 m | 800–1000 m |
|---|---|---|---|---|---|
| Large open pit mine | 10 | 8 | 4 | 2 | 0 |
| Small open pit mine | 10 | 4 | 2 | 1 | 0 |
| Large underground mine | 10 | 4 | 2 | 1 | 0 |
| Small underground mine | 10 | 2 | 1 | 0 | 0 |

The hydrological project data source and production process were the same as we used for assessing mining interference. We adopted the ecological impact assessment procedure for hydroelectric project construction developed for the Nuozhadu Nature Reserve [47] and set the maximum distance of influence at 5 km. For the construction of water reservoirs projects and hydropower development, the impact degree and scope of the project were evaluated according to the size of the project or the installed capacity. The criteria for classifying the size of hydropower stations were as follows: small-scale station under 5000 KW/h, medium-sized station of 5000–100,000 KW/h, large-scale station of 100,000–1 million KW/h, and mega-sized station over 1 million KW/h. Most hydropower stations were small and medium sized, with very few large hydropower stations and no mega hydropower stations. To obtain accurate vector data regarding the size of the areas impacted by water reservoirs and hydropower dams, this study assigned a width of 500 m from the center point of each project. From the circle described by this radius, 4 buffers of different widths were established as concentric circles at 1000 m, 1500 m, 2500 m, and 5000 m. The human influence scores assigned to each type of water reservoir and hydropower construction were based on our field investigation, relevant studies [47–49], and expert judgment, which accounted for the characteristics of each study area. Because of destruction caused by inundation, reservoirs behind dams were given the largest impact score of 10. This impact diminished gradually to zero approximately 2.5 km or 5 km upstream (Table 4).

China's multi-period Land Use and Cover Change data classified land use into 6 categories and 25 subcategories [50]. We assigned human interference scoring values for each subcategory of the Land Use and Cover Change data based on relevant studies [30,31] and expert judgment (Table 5). Shrubland and alpine grassland occur above the timberline, which begins generally at 3800 m above sea level [51,52]. The types of shrubs and grasses

that appear below 3800 m are mostly secondary types colonizing after the abandonment of land use [52] and were assigned human influence scores of 3 and 4. Grazing is common at altitudes above 3800 m, and grasslands were more disturbed by grazing than shrubland. Human influence scores of 2 and 1 were assigned to them, respectively. No new canals were excavated in the research area during the research period. Previously excavated canals were all modified by sand removal, quarrying, river dredging, etc. The canal mentioned in this study was either excavated or may have been a natural stream that was modified by human impact, and we assigned it a score of 2. Cropland, including dry land and paddy fields, was assigned scores of 7 in the practice of Venter et al. (2016) and Li et al. (2018) [31,53]. In mountain areas, paddy fields were consistently located in low elevations and dry land at higher elevations nearer the forest. Paddy fields were more disturbed by humans than dry land, and human influence scores of 7 and 5 were assigned to them, respectively.

**Table 4.** Parameter table of influence degree and scope of hydropower and water reservoirs facilities.

| Type | Dam Region | 500–1000 m | 1000–1500 m | 1500–2500 m | 2500–5000 m |
|---|---|---|---|---|---|
| large-scale hydraulic project | 10 | 8 | 4 | 2 | 1 |
| Small and medium hydraulic project | 10 | 4 | 2 | 1 | 0 |
| Large hydropower | 10 | 8 | 4 | 2 | 1 |
| Small and medium hydropower | 10 | 2 | 1 | 0 | 0 |

**Table 5.** The human influence scores assigned to each land use type based on relevant studies [30,31] and expert judgment.

| Land-Use Type | Sub-Type | Score |
|---|---|---|
| Cultivated Land | paddy field | 7 |
|  | dry land | 5 |
| Woodland | forested land (Altitude ≤ 3800) | 1 |
|  | forested land (Altitude > 3800) | 0 |
|  | shrubland (Altitude ≤ 3800) | 3 |
|  | shrubland (Altitude > 3800) | 1 |
|  | sparse forested land (Altitude ≤ 3800) | 2 |
|  | sparse forested land (Altitude > 3800) | 0 |
|  | other woodlands | 6 |
| Grassland | grassland (Altitude ≤ 3800) | 4 |
|  | grassland (Altitude > 3800) | 2 |
| Water Area | canal | 2 |
|  | reservoirs/ponds | 3 |
|  | Lake, permanent glacier snow area, tidal flat, floodplain | 0 |
| Industrial and Residential | urban land | 10 |
|  | rural residential area | 10 |
|  | other construction lands | 10 |
| Unused Land | desert, the Gobi Desert, saline-alkali land, swamp, bare land, bare rock and gravel fields, others | 0 |

### 2.2.2. Calculation of Ecological Intactness Scores

All data sets were computed with the fuzzy sum function (Function (1)) to obtain the comprehensive spatial distribution of Human Modification (HM) [30]. The ecological intactness score (EIS) was calculated on a 10-point scale, using an ArcGIS 10.5 grid calculator using EIS = 10-HM.

$$HM = 1.0 - \prod_{i=1}^{k}(1 - Fi/10) \qquad (1)$$

where HM is the score of all human interference, and $Fi$ is the score of the $i$-th type of interference.

$$EIS = 10(1.00 - HM) \qquad (2)$$

An ecosystem that is entirely natural and not at all impacted would have an EIS of 10.0, and one that was completely destroyed would have an EIS of 0.0.

### 2.2.3. Analysis of the Spatiotemporal Change Trend and Driving Force of EIS

The method of unary linear regression analysis combined with Theil–Sen median trend analysis was used to calculate the trend of EIS changes in the study area. The results of the trend analyses are coupled with the Mann–Kendall test of significance for the trends evident during 1980, 1990, 2000, 2010, 2018, and 2020 [54].

$$\text{Slope} = \frac{n \times \sum_{i=1}^{n} i \times SI_i - \sum_{i=1}^{n} i \sum_{i=1}^{n} SI_i}{n \times \sum_{i=1}^{n} i^2 - \left(\sum_{i=1}^{n} i\right)^2} \tag{3}$$

In the formula, Slope is the EIS trend slope of a certain grid in 1980, 1990, 2000, 2010, 2018, and 2020; $n$ is the total number of years; and $i$ is the annual ordinal. Slope > 0 indicates that the EIS trend increases, and Slope < 0 indicates that the EIS tends to decrease.

The Mann–Kendall test was used to determine whether the time series data had an upward or downward trend. The calculation formula is as follows:

$$Z = \begin{cases} \frac{S-1}{\sqrt{var(S)}}, S > 0 \\ 0, S = 0 \\ \frac{S+1}{\sqrt{var(S)}}, S < 0 \end{cases} \tag{4}$$

$$S = \sum_{i=1}^{n-1} \sum_{j=i+1}^{n} Sign\left(SI_j - SI_i\right) \tag{5}$$

$$\text{Var(S)} = \frac{n(n-1)(2n+5)}{18} \tag{6}$$

$$\text{Sign}\left(SI_j - SI_i\right) = \begin{cases} 1, SI_j - SI_i > 0 \\ 0, SI_j - SI_i = 0 \\ -1, SI_j - SI_i < 0 \end{cases} \tag{7}$$

We assumed that changes for the EIS were significant at the $\alpha = 0.05$ confidence level. We divided the test result $Z_C$ into significant change ($|Z_C| > 1.96$) and insignificant change $|Z_C| < 1.96$. Combining the analysis of slope and Z value, we classified the ecological intactness change trends into five categories: significant degradation, slight degradation, stable, slight improvement, and significant improvement (see Table 6).

**Table 6.** The classification standard of ecological intactness changes categories.

| Slope | Z Value | Category |
|---|---|---|
| ≥0.0005 | >1.96 or <−1.96 | significant improvement |
| ≥0.0005 | −1.96–1.96 | slight improvement |
| −0.0005–0.0005 | −1.96–1.96 | stable |
| <−0.0005 | >1.96 or <−1.96 | significant degradation |
| <−0.0005 | −1.96–1.96 | slight degradation |

We employed the Geodetector to analyze the driving force of EIS. The Geodetector is a statistical method to detect spatial stratified heterogeneity and reveal the driving factors behind it; this statistical method with no linear hypothesis has an elegant form and definite physical meaning [55,56].

### 2.2.4. Analysis of the Relationship between Ecological Intactness and Giant Panda Habitat Suitability

To understand the environmental characteristic of giant panda signs and the coupling relationship between ecological intactness and habitat suitability, we analyzed the habitat suitability of giant pandas within the study area. Currently, species distribution models (SDMs) are commonly used to predict the geographic range of a species [57,58]. As one

of the SDMs, the MaxEnt algorithm, was a proven powerful model when modeling rare species with narrow ranges and scarce presence-only occurrence data, and has been widely used in giant panda habitat assessment work [59–61].

This study employed MaxEnt models to evaluate the habitat suitability index (HSI) for the study area, with the habitat suitability index (HSI) ranging from 0 (unsuitable) to 1 (most suitable) [59]. The input variables included altitude, slope, vegetation type, annual average temperature, maximum yearly temperature, minimum yearly temperature, and annual rainfall [61]. We analyzed habitat suitability without considering human disturbance to avoid autocorrelation with the results of EIS. The parameters for model running were set as follows: we used 1066 giant panda signs, which came from the Fourth National Giant Panda Census, and selected 75% of the total 1066 presence data (n = 800) as training data, and the remaining 25% of presence data (n = 266) were used for testing the model. The habitat suitability index (HSI) map was obtained by the average training AUC (0.963) and test AUC (0.960), indicating that the performance of the model was reliable.

Next, we used linear regression analysis to analyze the correlations between EIS and HIS using the method of random point sampling [2]. Two types of points data were analyzed, which were giant panda signs that came from the Fourth National Giant Panda Census and 1000 random sample points from the study area. To further understand the ecological intactness characteristic of the giant panda habitat, we analyzed the EIS components by randomly sampling 1000 points from each type of giant panda realistic habitat, giant panda suitable habitat, and marginally suitable habitat, which were obtained from the Fourth National Giant Panda Census [62]. The technical framework of the study is shown in Figure 2.

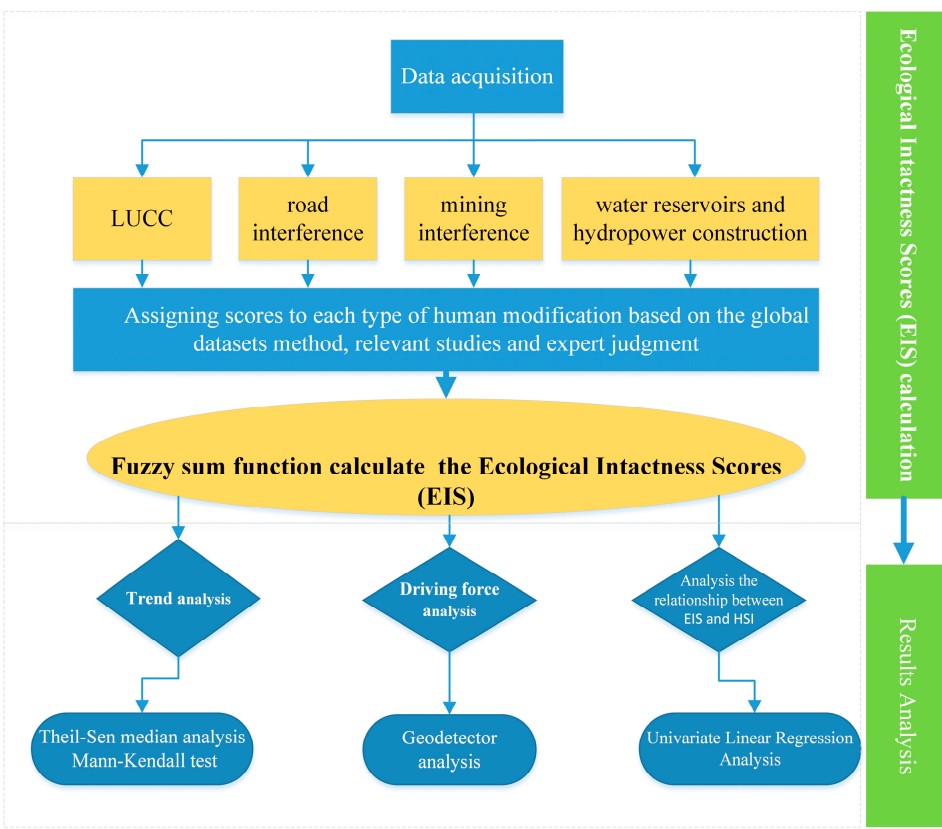

**Figure 2.** Flow chart of EIS calculation and technical framework of this study.

## 3. Results

### 3.1. Spatial Characteristics of EIS

In general, the study area has a high level of intactness (6.4 ± 2.2) (Figure 3), and the GPNP (7.1 ± 1.6) is more intact than exterior areas (6.0 ± 2.5). Most areas (92.6%) of the GPNP are above 6.0 (Figure 4A,B). One-fifth of the GPNP (about 4171.8 km$^2$) has an EIS of more than 9.0, which belongs to the intact ecosystem in the strict sense (Figure 5B).

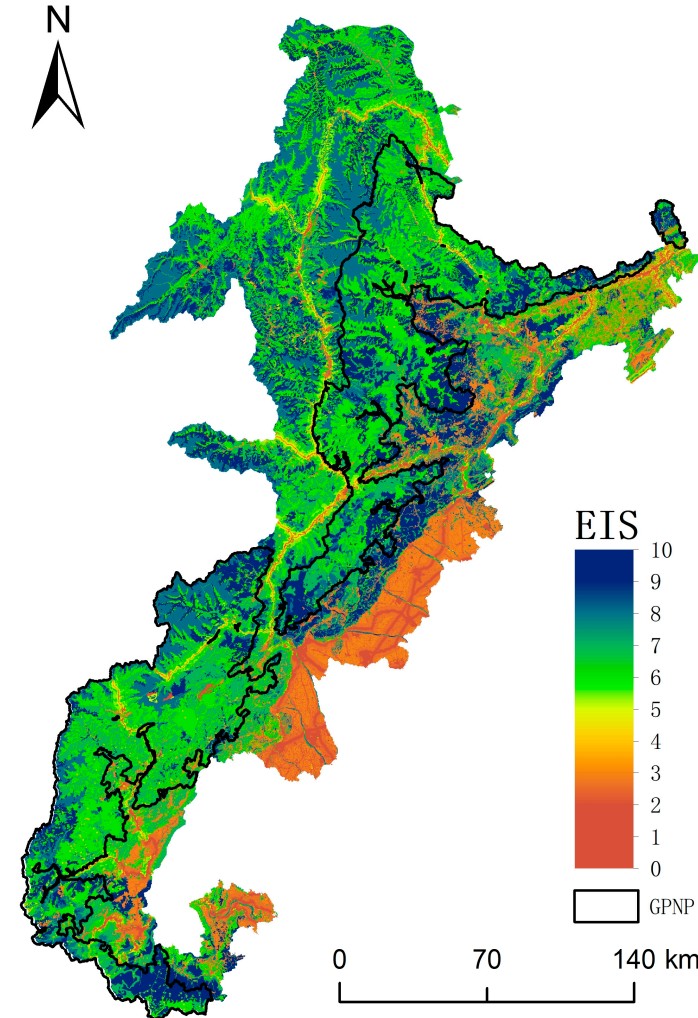

**Figure 3.** Spatial distribution pattern of different intactness levels in the study area in 2018.

In terms of spatial characteristics, from the comparison of mountain systems in the study area, the order of intactness levels is the mountains of Minshan (6.5 ± 2.2) > Daxiangling (6.3 ± 2.6) > Qionglaishan (6.1 ± 2.2) > Qinling (6.0 ± 2.3). The EIS in GPNP in the mountains of Qionglaishan, Minshan, Qinling, and Daxiangling was 6.9 ± 1.5, 7.2 ± 1.5, 7.1 ± 2.2, and 7.9 ± 1.5, respectively.

Within GPNP, the EIS in the east of Qionglaishan and south of Minshan was relatively lower than in other regions. Additionally, the ecological intactness scores in the north of Qionglaishan, south of Daxiangling, and middle of Minshan were relatively high (Figure 3).

Comparing the EIS levels of different altitude ranges, the mean EIS at elevations less than 1200 m was 6.0. The mean EIS values from 1200 m to 4000 m were between 6.5 and 7.5, and the EIS above 4000 m mean scores exceeded 7.5. Mean intactness scores below 1200 m in the GPNP were significantly higher than scores at the same altitudinal range in exterior areas (Figure 5A).

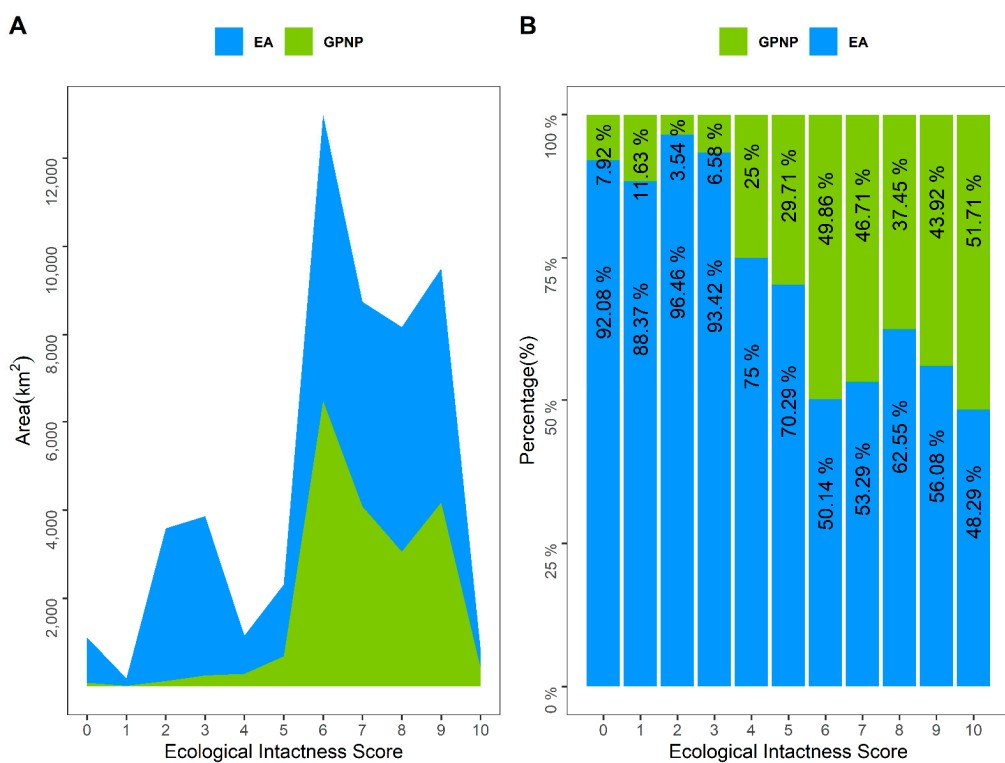

**Figure 4.** The area (**A**) and area proportion (**B**) of different ecological intactness scores within GPNP and exterior areas (EA) in the study area.

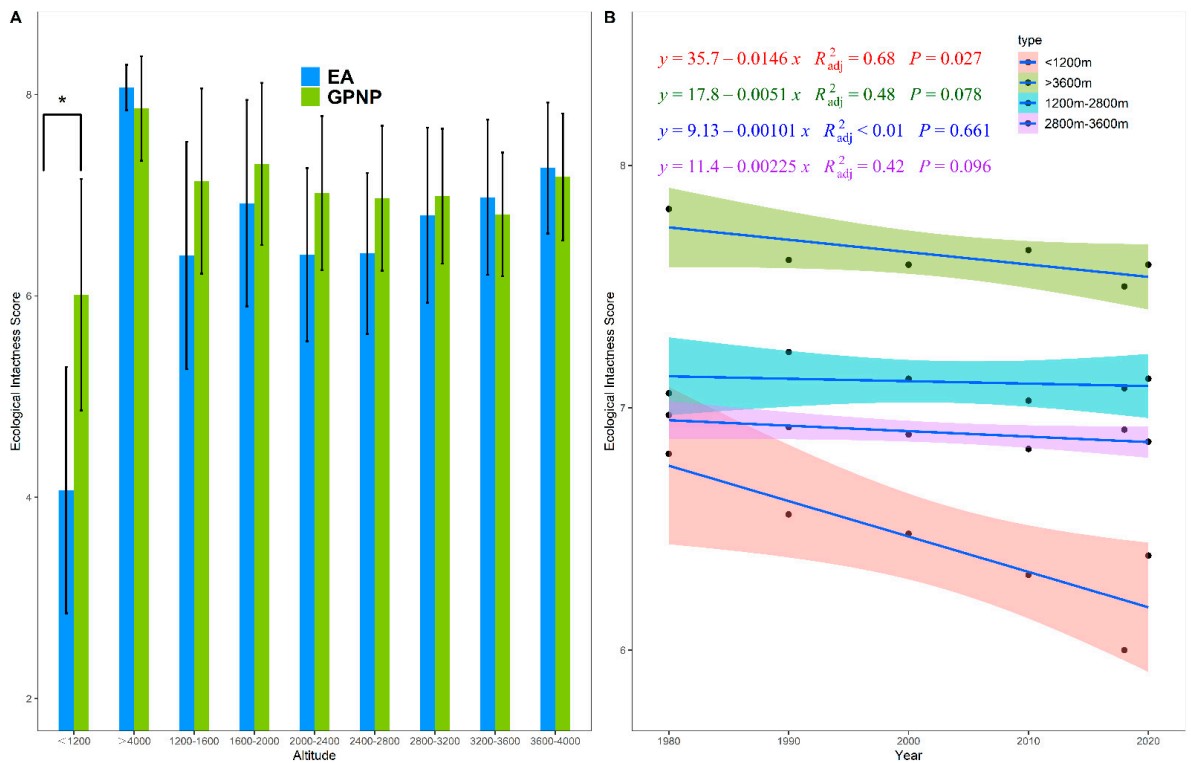

**Figure 5.** Ecological intactness score of different elevations between the GPNP and exterior areas (EA) for 2018 (**A**) and 40 years time-series dynamics for different elevations (**B**). * indicates a significant difference, and no mark indicates an insignificant difference.

*3.2. Fourty-Year Intactness Trends and Driving Force in GPNP*

The mean intactness scores in the GPNP from 1980 to 2020 are shown in Figure 6. The mean EIS for 1980, 1990, 2000, 2010, 2018, and 2020 were $7.2 \pm 1.4$, $7.2 \pm 1.4$, $7.1 \pm 1.4$, $7.1 \pm 1.5$, $7.1 \pm 1.5$, and $7.1 \pm 1.4$, respectively. Scores remained stable during the past 40 years in 79.88% of the GPNP (Figure 7). Ecological intactness underwent degradation in approximately 14.77% of the GPNP, of which about 0.99% (96 km$^2$) declined significantly. The areas showing a downward trend are mainly distributed along valleys and low-altitude areas (Figure 7). In 5.35% of the GPNP, intactness scores improved, and significantly so in 0.42% (82.79 km$^2$) of the park.

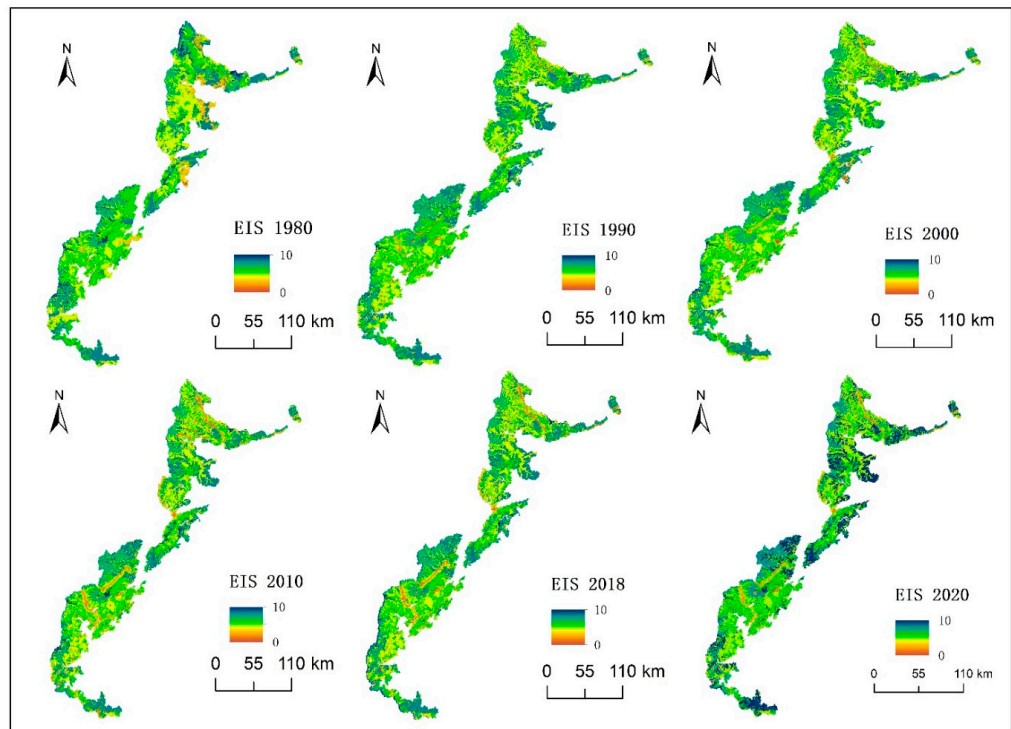

**Figure 6.** Characteristics of the ecological intactness change of GPNP from 1980 to 2020.

The mean value of ecological intactness below 1200 m was <7.0, and from 1980 to 2018 ecological intactness diminished (Figure 5B). EIS was stable at all elevations above 1200 m.

The driving force of ecological intactness change was analyzed by the geographic detector. The result shows that the explanatory rate of land use, road interference, mining interference, and water reservoirs and hydropower construction was 84.25%, 22.45%, 0.44%, and 0.017%, respectively. Land use and road penetration (not only road construction, but also for road usage) are the main factors affecting ecological intactness.

*3.3. Relationship between Ecological Intactness and Giant Panda Habitat Suitability Index (HSI)*

The correlation was not significant between ecological intactness scores (EIS) and the habitat suitability index (HIS), which means that a site with a high level of ecological intactness score does not mean a high level of habitat quality (Figure 8A,B). It can be seen from Figure 8A that the giant panda signs have a high level of suitability, and the linear correlation between the HSI and EIS at giant panda signs is not significant. Visual inspection of the 3D scatter plot is sufficient to detect not only the spatial distribution characteristics between the EIS and HIS, but also distribution differences concerning altitude (Figure 8C,D). Giant panda signs are mainly distributed in the areas with an EIS between 6.0 and 9.0. The realistic habitat of the giant panda is mainly distributed in the regions with an EIS above 6.0 (Figure 9).

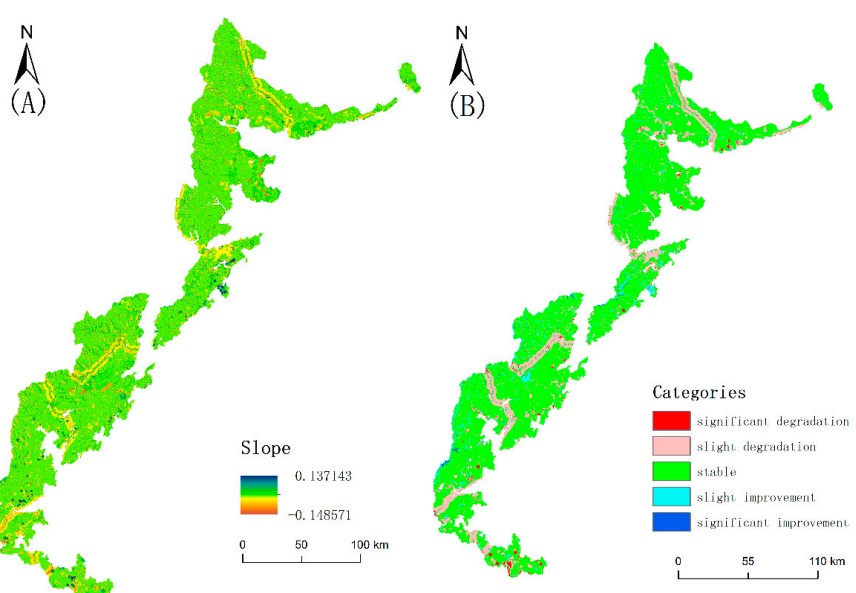

**Figure 7.** Change slope (**A**) and change category (**B**) of ecological intactness in the GPNP.

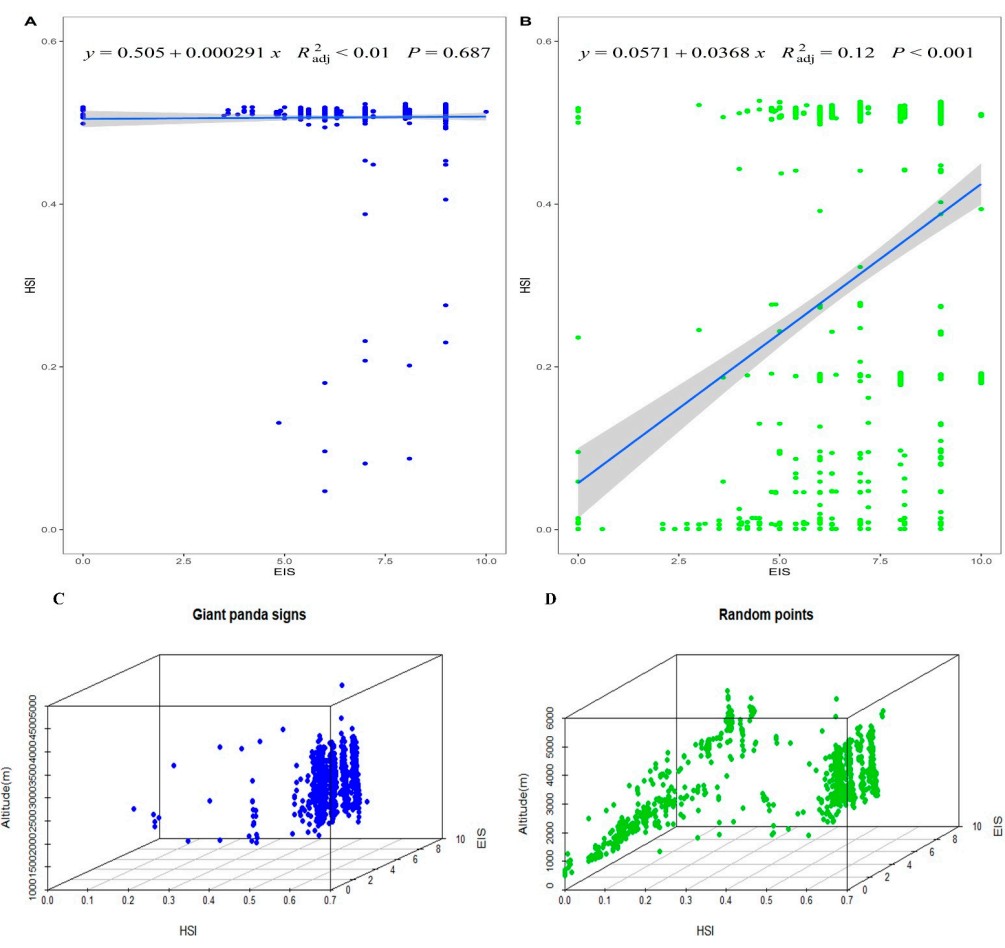

**Figure 8.** Linear regression analysis of the relationship between the ecological intactness score( EIS) and giant panda habitat suitability index (HSI) for giant panda signs (**A**) and random points (**B**); 3D spatial distribution relationship of altitude, ecological intactness score (EIS) and giant panda habitat suitability index (HSI) for giant panda signs (**C**) and random points (**D**).

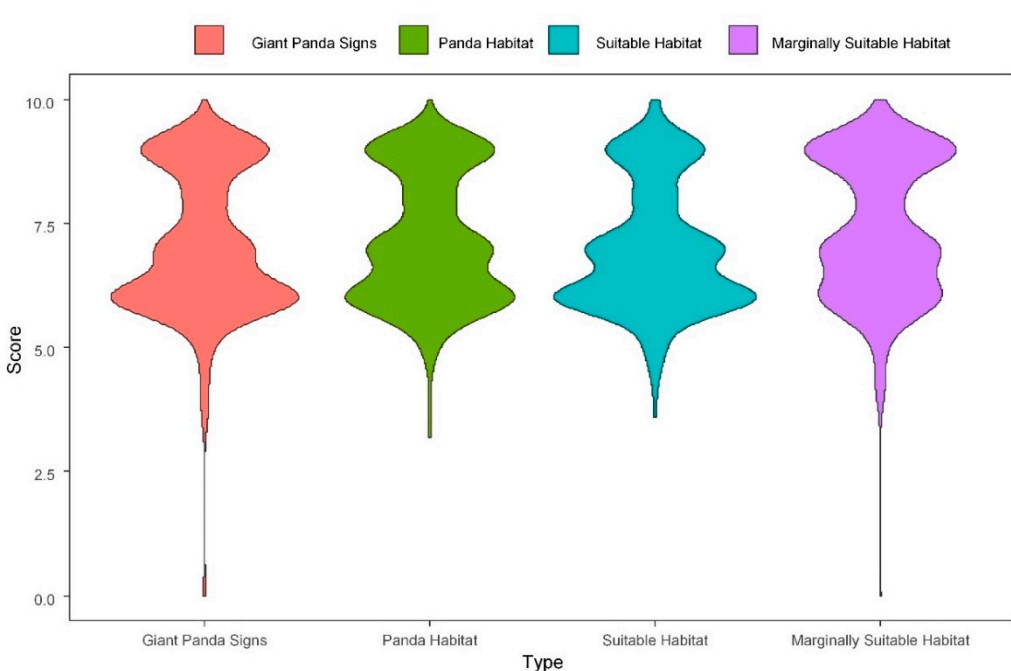

**Figure 9.** The ecological intactness characteristic of giant panda signs, giant panda realistic habitat, giant panda suitable habitat, and marginally suitable habitat.

## 4. Discussion

### 4.1. Ecological Intactness in Giant Panda National Park

This study reveals the ecological intactness status of giant panda habitat ecosystems and the spatio-temporal changes of human disturbance at the landscape scale. Sichuan GPNP consists of two parts, one part is the 26 former natural reserves comprising 10,985.36 km$^2$, and the other is the adjacent area to connect these former natural reserves, comprising about 9191.64 km$^2$. By comparing the EIS in the GPNP and exterior areas (EA), we found that the currently designated national parks have a higher level of ecological intactness than their exterior areas (EA). This result is not surprising because the earlier giant panda reserves were all included in national park management [63]. The GPNP has about 4171.8 km$^2$ (one-fifth of the GPNP) intact ecosystem, where the EIS is higher than 9.0. About 80% of the GPNP the ecological intactness has remained stable, but14% of the area has shown signs of slow degradation over the past 40 years. The degraded areas of the GPNP are mainly located in the adjacent area, which was not strictly managed in the early stage and has strong human activity. Systematic planning of ecological restoration and wildlife corridor construction should be carried out in this area.

EIS varies between mountain ranges in national parks; the EIS in the east of Qionglaishan and south of Minshan were relatively lower than in other regions. This part of GPNP is close to the populated area of Chengdu Plain and existing intensive human activities. On the other hand, the ecological intactness in the north of Qionglaishan, south of Daxiangling, and middle of Minshan were relatively high. These areas are the original giant panda nature reserves.

We further analyzed the status quo characteristics of regional ecological intactness in different altitude dimensions; EIS was especially low at elevation >1200 m, where human interference is intense. Human activities, such as hamlet construction and road construction, are mostly located in surrounding foothills and valleys, causing fragmentation of the panda habitat [4]. Li and others reported that free-ranging livestock threatened the long-term survival of giant pandas, which means that human activities in low-altitude areas, especially derivative disturbances, such as grazing and wild collection, drove giant pandas up to more elevated habitats [64–66].

This study has a close assessment result with global human modification (GHM) for the ecological intactness of the study area. The EIS result of GHM is twice that of the global human footprint (GHF), and the EIS of our study is in between. Differences in scale for resolution for buildings, arable land, night lights, and pastures (1000 m × 1000 m) were recorded by the global human footprint method [27,53], compared with the data for land use and coverage used in this study (30 m × 30 m), probably accounted for part of these discrepancies. GHF mapping of the area land cover types, which does not differentiate the intensity of the impact of those cover types (crop and pasture), may be the main reason for the difference [28].

*4.2. Ecological Intactness and Habitat Suitability for Giant Panda*

The GPNP takes ecological intactness as a conservation goal, whether or not the new goal can strengthen the protection of giant pandas and giant panda habitats. This study can give some positive information and strategies for this goal. Analysis shows that giant panda habitats and panda signs are not located in completely intact or natural areas; this does not mean that giant pandas do not like the natural habitat environment, but rather, the result reveals the broad impact of human activities on giant panda habitats. Because the suitable habitat for giant pandas is distributed on gentle slopes between 1100 m and 2800 m above sea level [44,67], these areas are also vulnerable to human activities such as roads, farming, and planting. Our result indicates that the EIS scores for the giant pandas' current habitat and their suitable habitats are at a relatively high level (>6); this level could be set as a threshold level for management of the human activity. It can also be used as a standard to adjust the boundaries of national parks; if so, most of the optimum and suitable habitats for pandas would be included in the protection of the GPNP.

From the spatial relationship between the ecological intactness score (EIS) and giant panda habitat suitability index (HSI), if a suitable EIS threshold is set, we can see that the aims to protect the ecological intactness may not only fulfill the habitat suitability goals, but also cover other giant panda habitats, and this is a potentially comprehensive conservation strategy.

*4.3. Application of EIS in National Parks Management*

As an example, this study evaluated the GPNP EIS baseline and measured the 40-year change. Through our case study in the GPNP, we suggest that ecological intactness will be an effective indicator for monitoring and assessing the impact of human activities on regional natural ecosystems. Protected areas with different human pressure baselines should establish different management measures to improve conservation effectiveness [68]. National parks need the establishment a comprehensive monitoring and evaluation framework including ecological intactness for managing human activities. Feng, Cao et al. developed a "baseline + change" framework to assess the effectiveness of protected areas [68]. This valuation framework of "baseline + change" could be taken in the National Parks monitoring system.

With the development of remote sensing techniques, fine-scale image was easy to obtain. The data resolution can shift to different levels to fit the management's needs. It is recommended to establish a land use management database in national parks combine with the periodic national land survey and to use high-resolution remote sensing or drone technology to achieve refined management of national parks. Managers can evaluate different scales by selecting images with suitable resolutions. Compared with traditional management indicators, such as forest coverage and forest volume, ecological intactness may be more comprehensive, and the monitoring costs will be lower.

Setting reasonable scientific goals and selecting appropriate evaluation protocols are key issues for the management of natural ecosystems preserves everywhere [6]. Otherwise, landscape patterns may change, and natural ecosystems may undergo undesirable modifications. National parks are the most strictly managed protected areas in China, and human activities are strictly restricted in this kind of area. Managers of the new GPNP are faced

with resolving many issues, among them the selection of ecological intactness thresholds for different managing zones. The GPNP is divided into two main zones including the core protection zone and the general control zone [69]. The EIS thresholds of the general control zone could be lower than the core protection zone, allowing for certain types of human activities such as ecological restoration, habitat improvement, and new corridor construction [63]. This idea of differentiated control target setting may help the effective carry out of post-2020 biodiversity conservation work.

## 5. Conclusions

This study provided an evaluation method combining remote sensing and field investigation to reveal ecological intactness. We synthesized a map of the ecological intactness, which spatially and intuitively showed the intensity of human activities in Giant Panda National Park and its adjacent areas. We assessed the current status and identified the historical changes from 1980 to 2020 of the ecological intactness caused by human impacts on the landscape scale. This study found that Giant Panda National Park has a high ecological intactness baseline but still can be improved. Low-elevation areas (<1200 m) where a significant decrease occurred should be regarded as the focus of conservation management. From the mountain system point of view, the ecological intactness in the north of Qionglaishan, south of Daxiangling, and middle of Minshan were relatively higher than east of Qionglaishan and south of Minshan. During the past 40 years, the ecological intactness in most areas of the GPNP remained stable, and about 14% of the area was degraded. LUCC and road construction were the main driving factors for the decrease of ecological intactness in the GPNP. The habitat of the giant panda is mainly distributed in the regions with an EIS above 6; this is the key link between ecological intactness and habitat suitability. Our research recommends that the ecological intactness score (EIS) can be used as one of the evaluation and monitoring indicators for the protection effectiveness of the national park.

**Author Contributions:** Conceptualization, C.L. and P.L.; methodology, C.L. and H.Y.; software, C.L.; validation, C.L. and H.Y.; formal analysis, C.L.; investigation, C.L., X.W., J.W., H.L., C.M., L.M., H.J., S.W., Y.C., Y.H. and W.X.; writing—original draft preparation, C.L.; writing—review and editing, P.L. and S.L.; visualization, C.L.; supervision, P.L.; project administration, P.L.; funding acquisition, P.L. All authors have read and agreed to the published version of the manuscript.

**Funding:** This research was funded by the National Key R&D Program of China (2016YFC0503305), Management Framework and Capability Building for Development of Ya'an Giant Panda National Park (NOR/15/301/16/002), and the Sichuan Science and Technology Program of Key Technology and Demonstration for Biodiversity Conservation in Giant Panda National Park, Grant/Award Number: 2018SZDZX0036.

**Institutional Review Board Statement:** Not applicable.

**Informed Consent Statement:** Not applicable.

**Data Availability Statement:** The data presented in this study are available in the article.

**Acknowledgments:** Thanks to Andre F. Clewell for his contribution to improving the language of the manuscript. We also thank Shaoyao Zhang and Guyue Hu for their help in data acquisition and analysis for this work.

**Conflicts of Interest:** The authors declare no conflict of interest.

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
