# Peer review of "Spatial-Temporal Change for Ecological Intactness of Giant Panda National Park and Its Adjacent Areas in Sichuan Province, China"

_diversity, doi:10.3390/d14060485_

Round 1

Reviewer 1 Report

The study is interesting and methods used are appropriate.

The manuscript would benefit with a careful review as there are many typos, punctuation problems and misused worlds. Also tables and figures descriptions must be improved.

I also made a comment regarding naturalness that could be a good addition to the rationale.

L31- Delete “There are some key findings from this study”.

L43-45 – Does GPNP is within the hotpots defined by Mittermeier et al. (2000). If so, please add the reference. If it does not, the sentence must be revised as it induces the reader to think it does.

Mittermeier RA, Myers N, Gill PC, Mittermeier CG. 2000. Hotspots: Earth’s Richest and Most Endangered Terrestrrial Ecoregions. Mexico City: CEMEX.

L50-51 – It would be beneficial to the manuscript if authors add the concept of naturalness and how it could be useful in the presented context. There are several publication on the subject. A few examples:

WINTER, S. Forest naturalness assessment as a component of biodiversity monitoring and conservation management. Forestry, 2012. Disponível em: <http://forestry.oxfordjournals.org/content/early/2012/02/29/forestry.cps004.abstract>.

An extensive review on the topic can be found here: https://haydon4.tripod.com/index.htm

L54 – I would be very careful if the assumption that integrity is “the degree to which an ecosystem retains its natural state unmodified by human activity” as basically no ecosystem in the world is free of human influence. It is even more important if one is assessing the absence of human interference by remote sensing in order to find changes or disturbances as in the present case.

L60-78- Following the previous rationale, the search for a non-disturbed area is unachievable and the evidence for human-driven changes probably requires on the ground research. Authors must consider those aspects in their manuscript.

L99-100 – “Taking the regional characteristics of GPNP and data 99 availability into consideration.” The sentence is lost here.

L100-103 – This part seems to be better off in methods section.

L104- “aimed at” .

L107-108 – The question raised by the authors – the use of HSI -  is very important and it was taken into consideration in the analysis although it is not clear in the text. Please improve.

L116-117 – Revise English: counties do not “occur” in a park.

L125 – replace “persons” for “people” or “inhabitants”.

L131 – why logging is not used as one of the layers/threats for evaluating integrity of GPNP?

F134 – Improve Fig 1 description. Mention that the top inlet is China, etc. Also, what is “county” in the map?

L137-138 – Describe what is inside the table instead of simply calling the Table.

L 139-182-190 – Who is the “expert”? Judgment or opinion/expertise?

L141-142 – Delete “Table 2 presents 141 road grade influences” and add its information onto the previous sentence.

L149-151 – First sentence contradicts the second. Or authors mean that data is not available online.

L151 – Replace purples for purpose.

L157 - according to or in accordance with the. Revise.

L160 – Denude is not a correct word.

L176- Replace “giant” for mega as a matter of consistency.

L177- Please revise the use of “installation” here and after.

 Table 5 – Revise symbols that should be commas.

L276- Delete sentence and add reference to Figure 3 to the following sentence.

L277-282 – The text is very confusing with so many numbers. Please call attention to trends and cite specific numbers in brackets. I would also eliminate the second decimals in all values.

Fig 4 and 5 – Improve description including also what A and B mean.

Fig 6- I suggest moving the year from below to North to the legend (beside EIS e.g. EIS 2020).

L316 – Delete sentence and add reference to Figure 5 onto the following sentence.

L326 – If authors found no correlation, it should be “not significant” instead of “insignificant” .

Reviewer 2 Report

The idea of the manuscript is very interesting, and it is well in line with the scope of the journal. Studies like this one are important and needed in China and global conservation context. To improve the overall quality of the manuscript, I have some suggestion/comments as below:

 (1) Abstract: The abstract need to be overall improved.

 (2) Introduction:

      - The global and national background about “ecological intactness” should be given in the first paragraph, which is helpful bring your study into a more general topic.

    - In introduction section, author addressed method of human footprint and human modification, and author described many detail information. This section should be improved by substantially review the theory and method of “ecological intactness” and its driving force studies.

(3) Materials and Methods

   -It is better provide GPNP location, longitudeand latitude information in the study area to help international reader better understand.

   -The legend of figure 1 should be improve with same font and size. The location name in the figure should clear.

   -In the formula 3, “Slope” refers to trend of EIS changes. Why use “slope”? Is it a widely used formula called “Slope”? if not, it is better to use other name to avoid the ambiguity.

    -The habitat suitability index(HSI) map obtained AUC(0.963) and test AUC(0.960) should be cautions the possibility of model overfitting.

 -The figure could be better organized with better expression.

(4)  Results:

  - Spatial characteristics of EIS Please describe the main result and findings. This part need a concise 

  -Relationship between Ecological intactness and giant panda habitat suitability index (HSI) In this section, author tried to explore the relationship between Ecological intactness and giant panda habitat suitability index (HSI). Please clarity the main finds from Fig.8 Linear Regression Analysis.

  -The habitat of the giant panda mainly distributed in the regions with an EIS above 6 is an interesting finding. Please highlight in your abstract and conclusion.

  -There are 9 figures in this paper. Author could put some figures in the supplementary.

 (5) Discussion

  -The discussion could be improved. Please compare the main findings with previous studies and discuss these main points.

  -The value of this paper is not only for giving implications on GPNP, but also for the ambitious national park plan of China and post 2020 global conservation goal. Please discussion the implications in the China’ national park system and global conservation context.
